# CTM and QFD analysis: Framework for fintech adoption priority in commercial banks

**Donghoon Kang** [1], **So Young Sohn** [2]*

1 Graduate Program in Technology Policy, Yonsei University, Seoul, Republic of Korea, 2 Department of Industrial Engineering, Yonsei University, Seoul, Republic of Korea

* sohns@yonsei.ac.kr

**Data Availability Statement:** Data cannot be shared publicly because of license issues. The patent data we used is the "patstat" database. The "patstat" database is a patent database provided by the European Patent Office. You can access patstat

## Abstract

As financial technology (fintech) is developing rapidly, many commercial banks experience difficulty deciding what kind of fintech to primarily focus on when managing their business. Owing to limited resources and assets, there is a practical need for guidelines for banks' investments in fintech. This study provides a systemic procedure to identify promising fintech groups and their investment priorities. We propose a QFD-based decision support framework for banks by considering both aspects of the emerging fintech push identified using patent topic modeling and the market pull of banking services obtained from a survey of the literature and experts. An empirical application of the proposed QFD framework to major South Korean banks shows that transaction support technology, secure transactions, and trading platforms are the three most important fintech categories. The QFD results are utilized to guide individual banks for further investment strategies such as mergers and acquisitions, strategic partnerships, and spin-off operations. The proposed framework can be generalized and applied to other financial service firms.

## 1. Introduction

As digital transformation technology is evolving, the banking sector has been actively adapting to advanced financial technology (fintech) to offer new banking products and services and gain competitiveness for better market positions. The literature notes that technological innovation in the banking sector influences numerous services. From a banking system perspective, fintech has been undergoing intensive discussions following the rapid growth of patents related to machine learning, blockchain, and robot advisers [1,2]. Fintech-driven innovation has gained attention for its complementary effect on traditional commercial banks, which already have a large amount of data related to customer information, including customer transaction behavior [3]. Big data, which are owned by traditional commercial banks, are highly accurate, complete, and reliable, and provide a unique advantage because they can be utilized to reduce credit risk and predict borrowers' behavior [4]. Commercial banks, as a backbone of the banking system, are advised to integrate modern technologies to retain users and remain competitive [5,6]. Major global banks, such as the Bank of America, JP Morgan, and Wells Fargo, hold many patented technologies. Nevertheless, these banks have not created

via "epo.org//patstat". Other relevant data are within the manuscript.

**Funding:** The corresponding author received the National Research Foundation of Korea (NRF) grant funded by the Korea government (MSIT) [grant number 2020R1A2C2005026]. The funders had no role in study design, data collection and analysis, decision to publish, or preparation of the manuscript.

**Competing interests:** The authors have declared that no competing interests exist.

innovative new business models using these technologies. This indicates that integrating new technologies and services is more complex than simply developing technology.

The challenges facing Korean banks are more threatening than those of global banks. Mobile applications created by Fintech-based startups with large-scale investments have secured monthly active users. They exceed those of commercial bank mobile apps in a short period of time, and the gap is gradually widening [7]. A greater threat is the emergence of Internet-based banks. In particular, the rapid asset growth of Kakao Bank, a subsidiary of Kakao, which controls more than 95% of the Korean mobile messenger market, could be a factor in declining market share of banking industry. Korean commercial banks that lack IT-based technology and manpower are in a rush to imitate the rapid service launches with modern technologies of fintech-based startups and internet-only banks [8]. Therefore, Korean commercial banks are aware that it is essential to secure fintech to become a leader, not a follower. However, as can be seen from the examples of major global banks that have sufficient technology, more complex considerations are needed because simply securing technology does not create nor guarantee a new business model.

The complexity of integrating new technologies and services leads to potential ambiguity for many banking firms in setting up their plans to employ fintech to transform financial services to those that are cost-effective, convenient, reliable, and meet customer needs [9]. Rooted in the technology adoption theories, the adoption of fintech is yet to be realized due to the remaining concerns of users regarding fraud, data protection and cyber-attacks [10]. In such situations, practical guidelines for configuring fintech priorities should be employed. Specifically, a decision-making support framework is necessary for banks that consider the aspects of the fintech push of related emerging technologies and the market pull of banking services. However, most fintech or digital technology-related studies for banks have concentrated on verifying the positive impact of digital finance in stimulating innovation [11], diffusing mobile-based branchless banking services [12], encouraging user participation in risky financial markets [13], identifying failure factors of banking information systems [14], and gaining users' trust and adoption intention of Internet banking [15].

Different technologies may be required to improve the services offered by banks. For example, to refine the loan interest rate, it is necessary to accurately estimate the default rate of borrowers. Deep learning and text-mining algorithms can be used for this purpose. Conversely, a specific technology can be applied to multiple services. For instance, deep learning can be used to estimate the corporate default rate, but it can also be used to recommend an appropriate portfolio to customers. While recent waves of digital innovations have led to a positive outlook for various fintech application projects, traditional and commercial bank managers and executives face challenging tasks in prioritizing investments and developments [16]. Therefore, it is necessary to understand the interrelationship between the various services that banks offer and the different types of fintech applications that are available.

Based on the aforementioned challenges, we provide a systemic procedure that can be used to identify promising fintech groups and their investment priorities. Our proposed Quality function deployment (QFD)-based decision support framework considers the fintech push and market pull of banking services. To consider the market pull aspect, we classified the services provided by commercial banks through the literature and expert surveys. For the fintech push, contextualized subject modeling is applied to patents to identify emerging fintech groups. We then applied the proposed framework to patent data filed at the United States Patent and Trademark Office (USPTO) with a survey of major South Korean commercial banks. Subsequently, fintech investment strategies are proposed based on the empirical results. This study uniquely contributes to banks' efficiency in terms of fintech adoption by proposing an approach that fulfills research and practical needs.

The remainder of this paper is organized as follows. Section 2 provides a literature review of traditional commercial banks' perspectives on fintech. Section 3 provides an overview of the research methodology of QFD along with contextualized topic modeling (CTM) analyses and the survey of major South Korean commercial banks. The analyses results, implications, and the investment strategies for fintech adoption by banks are discussed in Section 4. Lastly, the conclusions are discussed in Section 5.

## 2. Literature review

This section reviews research on fintech adoption and the relationship between innovation and performance from the perspective of commercial banks.

### 2.1. Choices of fintech adoption

Fintech and its adoption in the banking industry have brought innovations in various financial services, such as credit services, deposit services, financial market trading and brokerage, financial product advisory and retail sales, and transfer and global remittance [17–21]. Based on the predicated positive influence of fintech adoption in the commercial banking business model, various suggestions have been made regarding the scope of choices. For example, Thakor [17] proposed the following four services for fintech innovation: (i) credit, deposit, and capital-raising services (i.e., crowdfunding, lending marketplaces, mobile banks, and credit scoring); (ii) payments, clearing, and settlement services (i.e., mobile wallets and digital exchange); (iii) investment management services (i.e., e-trading and robo-advice); and (iv) insurance services (i.e., data-driven risk pricing and contracts).

However, the advent of fintech can also provide an unexpected challenge in sustaining market demand for commercial banks with traditional business models focusing on the traditional financial market. Grobys et al. [22] found that fintech-embedded lending services can improve financial intermediation in mortgage markets. While traditional banks generally provide lending services that charge minorities higher fees for purchase and refinance mortgages, the recently proposed fintech-embedded service can significantly reduce potential discrimination using algorithms compared with in-person services [23]. Baker and Wurgler [24] also noted that independent mobile payments can lower overall costs by utilizing cloud computing to store and manage user data efficiently and, ultimately, offer faster payment processes.

Despite breakthroughs in technology development, traditional commercial banks' conservative and less strategic approaches to technology adoption have suffered from losing new market opportunities to fintech startups or new market entrants. Bunnell et al. [25] noted that implementing fintech should lead to potential solutions to the challenges faced by traditional financial advisory services, thereby ensuring improved services from both the service provider and user perspectives.

Thus, commercial banks with traditional business models must prioritize fintech applications based on market demand, target services, and patent-based technology readiness.

### 2.2. Strategies for fintech adoption

The general adoption or investment choice for which financial services should be improved depends on technology readiness and service strategies. Therefore, commercial banks must consider appropriate investment strategies for a desirable outcome given the intended scope of the services. Recent trends in commercial banks' efforts to acquire and develop intellectual capital can be explained from two perspectives: internal efforts (i.e., hiring data scientists and operating internal projects for innovation) and external efforts (i.e., funding or participating in joint ventures or mergers and acquisition of technology companies). Brandl and Hornuf

[26] differentiated investments from three perspectives: full integration of another company, strategic partnership between firms, and spin-off operations led by banks with traditional business models. They identified that banks with traditional business models must consider the possibilities of technology-driven digitalized financial services and coordinate common technological standards and banking functions to realize appropriate performance.

Intellectual capital includes intangible elements, such as knowledge, skill, information, and organizational structure, and tangible elements, such as patents, licenses, trademarks, and trade secrets [27]. IT investments are often a unique investment in acquiring intellectual capital in the banking industry [28]. The banking industry heavily invests in new technologies to satisfy service users' expectations and improve their overall experience [29]. Wang et al. [28] noted that small banks' tendency to overinvest and large banks' tendency to underinvest in technological development negatively or insignificantly impacts intellectual capital. The degree of investment in digital transformation or IT infrastructure requires strategic planning to achieve desirable performance.

Investment planning and strategic approaches have been proposed in several studies. Daim et al. [9] adopted patent co-citation analysis to better understand the emerging Internet of Things (IoT), cybersecurity, and blockchain technologies. They noted that the strategy of patent layouts and the development speed of innovative technologies are considered critical elements in determining the overall performance of IoT, blockchain, and cybersecurity. Baumann et al. [30] utilized patent documents to determine which countries invest in specific technologies and identify potential innovation trends for energy technologies. They suggest that firms use strategic patenting to demonstrate their technological strategy for marketing purposes. Furthermore, they noted that the suggested analytical approach could provide information on patenting strategies between national and international patenting activities. Lastly, Duho and Onumah [31] emphasized the critical role of decision support units in achieving investment efficiency, as they positively drive intellectual capital performance.

## 2.3. Relationship between technological innovation and performance

Theoretical foundations of technology adoption research share two viewpoints: adopters (user-level) and service provider (firm-level). The most prominent theories include theory acceptance model (TAM), unified theory of acceptance and user of technology (UTAUT), diffusion of innovation (DOI), and dynamic capability to name a few [32–35]. Among the theories, firm-level viewpoint often relies on DOI and dynamic capability [35,36]. Similarly, financial service firms (including both traditional and internet-only banks) anticipate the growth of capabilities leading to the innovation outcome and the potential performance improvement [37]. The dynamic capability view elaborates how firms can utilize capabilities and the external trend to gain competitive advantage in the market [38]. Based on the view, the technology adoption in financial services can be regarded as a modification or a complete renewal of existing service capabilities to fulfill the market's needs.

Technology-driven financial service development approaches produce innovative outcomes for new business models, applications, and process or products [39]. Consequently, commercial banks initiate and accelerate various types of research and development activities for patent acquisition. For example, Wang et al. [40] suggested that the levels of fintech innovation outcomes (reduction in bank operating costs, service efficiency improvement, strengthened risk control capabilities, and enhanced customer-oriented business models) depend on the bank's use of technological innovation. Wang et al. [28] empirically validated that a positive impact of IT investments on intellectual capital can lead to competitive advantage, contingent on firm type, size, positioning, and location.

However, because of the nature of patents which take time to come into effect, commercial banks often need government support and selective investments for desirable outcomes. For example, investment-driven technological innovations require time to actualize and may not show immediate results at initial financing [41]. Investments in internal projects alone do not directly lead to investment performance, and government regulations must be considered for a fintech boom to become apparent [42]. Haddad and Hornuf [43] noted that the availability of Internet server security, mobile subscription, and labor force also affect the development of fintech-driven markets. Moreover, the diffusion of financial service platform is often affected by the adoption intentions of the user groups [44]. Based on various efforts to understand the relationship between the development of intellectual capital (such as patents), banking competitiveness, and performance, internal and external factors must be considered [28].

There remains a critical decision support question regarding the financial services that technology must be strategically applied to in an orderly fashion to appropriately fulfill target customer needs from commercial banks anticipating the digital transformation of traditional business models.

## 3. Research methodology and data analysis

Based on a thorough literature survey, we propose a patent-based QFD framework by first identifying the areas for financial services with sublevels. Second, emerging technologies in the banking industry are classified into several areas by applying CTM to the abstracts of fintech-related patents. QFD is applied to identify the priorities of emerging fintech areas concerning the prioritized needs of the financial service categories. Empirical results were obtained by applying the proposed framework to patent data filed at the USPTO, along with a survey of major South Korean commercial banks. Fintech employment strategies are proposed based on this analysis.

QFD is a systematic framework originally developed for enhancing overall product and service design qualities by setting design targets based on the user's needs and requirements [45]. The QFD application has proven useful in engineering and management for its usefulness in resolving design improvement solutions from a *what* and *how* perspective [46]. Specifically, for a complex problem related to technology development trends and changing dynamics in service characteristics, QFD can simplify decision-making problems. QFD has been applied to identify emerging robot technologies [46], innovative services in the healthcare industry [45], and technology implementation orders [37], thereby proving its usefulness and applicability in forecasting technologies.

As case companies for commercial banks, four of the largest banks ranked by asset value in South Korea were utilized in this study. Namely, Woori, Shinhan, KB, and Hana financial groups are the only nationwide companies headquartered in South Korea. There are 19 commercial banks in Korea. Among them, 5 are special purpose banks owned by the government, 6 are local banks that can operate only in the provinces, 3 are Internet based banks, and 2 are foreign banks headquartered overseas. The four banks selected for the study are privately owned, nationally operational banks. In other words, only the four banks selected in the study are commercial banks that operate freely without any particular purpose. These firms have approximately over 305 billion USD (400 trillion KRW) total assets with over 25,000 employees. They represent traditional and commercial banks in South Korea for their historic establishment, going back to the early 1900s, with hundreds of branches in South Korea and global branches in other countries. These firms offer retail, corporate, and international banking; credit card operations; foreign exchange; and other services. Other commercial banks with different headquarters but available in South Korea are excluded from the study for the accuracy of the classification process by field practitioners (Section 3.1).

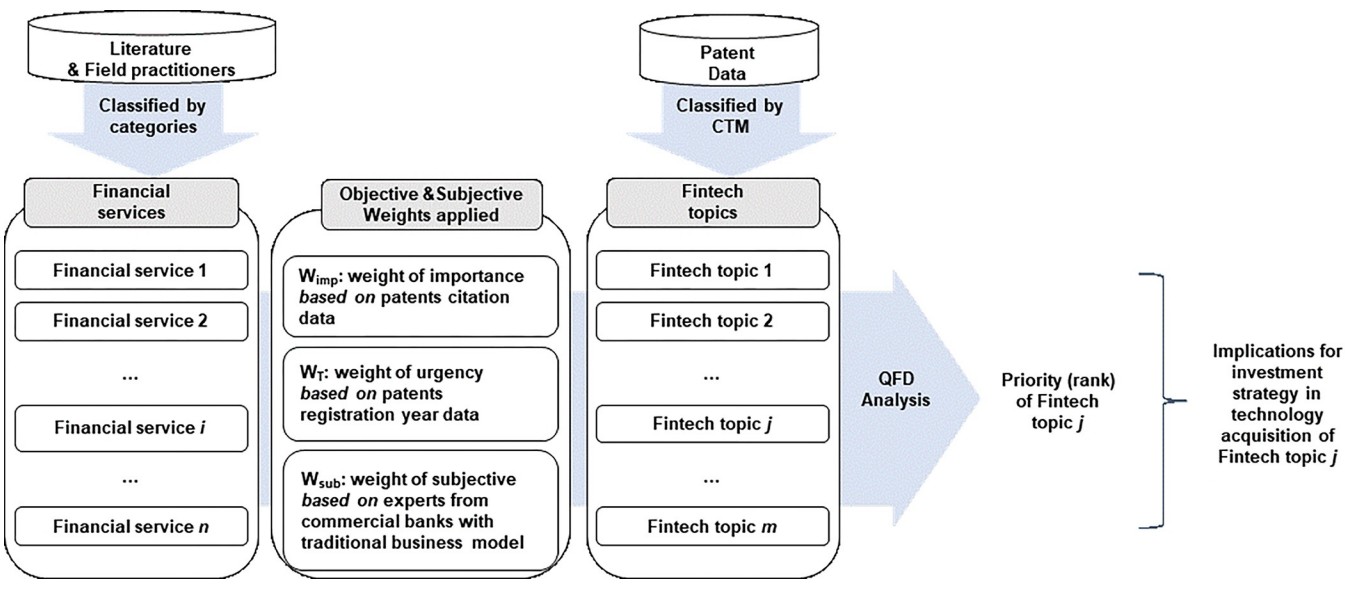

**Fig 1. The proposed framework.**

**Fig 1** illustrates the derivation procedure of the fintech priorities for commercial banks using QFD framework. First, we describe how we classify the financial services of commercial banks. Second, we explain how the banking industry's fintech technology can be divided using CTM. Third, we apply the QFD methodology to derive "fintech priority" based on finance-related patent data and opinions collected from commercial bank practitioners.

## 3.1 Identification of financial services of commercial banks (WHAT)

As displayed in **Table 1**, five distinct types of financial services are categorized, followed by examples of details of services that are provided, impacts of technology, and keywords that represent financial services well. This information was obtained from the perspective of commercial banks.

## 3.2 Classification of fintech categories (HOW)

This study classifies fintech using the latest topic modeling technique, CTM analysis. CTM provides keywords in each classification cluster (or topic cluster), providing baseline data to understand technology characteristics, such as readiness for application in financial services. This study utilizes patent data to identify technical characteristics or classifications to create a relationship matrix. Utilizing US patent data and the collection of abstracts, 12 topics were extracted by applying CTM. The details of each step are as follows:

*Step 1) Collection of patents using International Patent Classification (IPC) code*

Concerning development and patent registrations, the USPTO has been gaining attention for its leadership in the global trend [48]. In a recent study by Liu and Qiao [49], the USPTO demonstrated a significant degree of leadership in the patent subject and proportion of profitable patents. Therefore, we collected patents filed at USPTO, an appropriate representation of the fintech development trend, with IPC codes G06Q 20 or G06Q 40 from 1972 to 2020, as displayed in **Fig 2**. We used the G06Q 20 and G06Q 40 because they directly relate to the financial system. The former concerns "payment architectures, schemes or protocols," and the latter concerns "finance; insurance; tax strategies; processing of corporate or income taxes." We

**Table 1. Classification of financial services based on the literature.**

| Core financial services | Examples of services | Examples of technology disruption | | References |
|---|---|---|---|---|
| Credit Services ($i = 1$) | • Credit loan• Mortgage loan• Credit guarantee• Mobile banks, credit scoring | • Mobile credit loan<br>• Crowdfunding<br>• P2P lending | • Loan,<br>• Lending<br>• Credit<br>• Guarantee | [17–20] |
| Deposit Services ($i = 2$) | • Deposit or installment savings• Digital currencies• Mobile wallets | • Smart contracts in trade services and lending<br>• P2P lending | • Deposit<br>• Saving | [17–20] |
| Financial Market Trading & Brokerage ($i = 3$) | • High-frequency trading• Copy trading• E-trading• Individual or group financial investments brokerage | • Social trading<br>• Online brokerage | • Brokerage<br>• Sales<br>• Trading<br>• Platform<br>• Investment<br>• Pricing | [17,18,20,21] |
| Financial Product Advisory & Retail Sales ($i = 4$) | • Private banking services• Asset allocation advisory service• Trust services• Robo-advice | • Robo-advisory services<br>• Branchless banking services | • Trust<br>• Advisory<br>• Advice | [17,18,20,47] |
| Transfer & Global Remittance ($i = 5$) | • Letter of credit services<br>• International trade settlements<br>• Global remittances<br>• Peer-to-peer transfer | • Global peer-to-peer money transfer of money in different countries<br>• Blockchain-based markets<br>• Foreign exchange applications | • Transfer<br>• Remittance<br>• Letter of credit<br>• SWIFT | [18,20,47] |

conducted research on patents filed after 2011 because most of the patents have been applied for since 2011 (the proportion of patents filed after 2011 is 90%). In addition, patents prior to 2011 were excluded as they were too outdated for use in this study using the latest fintech.

*Step 2) Extraction of topics*

CTM was performed based on the abstract of the patent using Python. Before applying topic modeling, punctuation and insignificant words were removed. Specifically, we removed "stopword," which frequently appears in sentences but contributes little to semantic analysis. A graphical analysis method was used to obtain the appropriate number of topics. Using the dimension reduction method (PCA) and keyword extraction, we visually extracted the distance by topic and the number of overlapping classified topics. Using this approach, the number of topics with minimal overlap was selected while changing the number of topics from 10 to 20. Through this process, we set the number of topics to 12. Owing to the CTM, we obtained relevant topic words for the 12 topics, along with the probability that each patent belongs to a specific topic. **Table 2** lists the 12 topics and relevant topic words.

*Step 3) Overview of anticipated technology-driven outcomes*

To classify each topic as a technology, 10 patents with the highest probability of being included in each topic were selected. The abstracts of the corresponding patents were carefully read and analyzed. Subsequently, each topic was classified based on technology by comprehensively considering the relevant topic words and technologies included when reading the abstract of a patent (see **S1 Appendix**; The availability of detailed datasets can be discussed upon request under the consideration of license status.).

## 3.3 QFD analysis

QFD is a systematic framework that can be utilized to prioritize the order of investment in varying fintech. The framework consists of a matrix-like structure and provides decision-making support for identifying the relationship between customer requirements (i.e., financial services) and technical solutions (i.e., fintech topics). Using the interrelationship between the

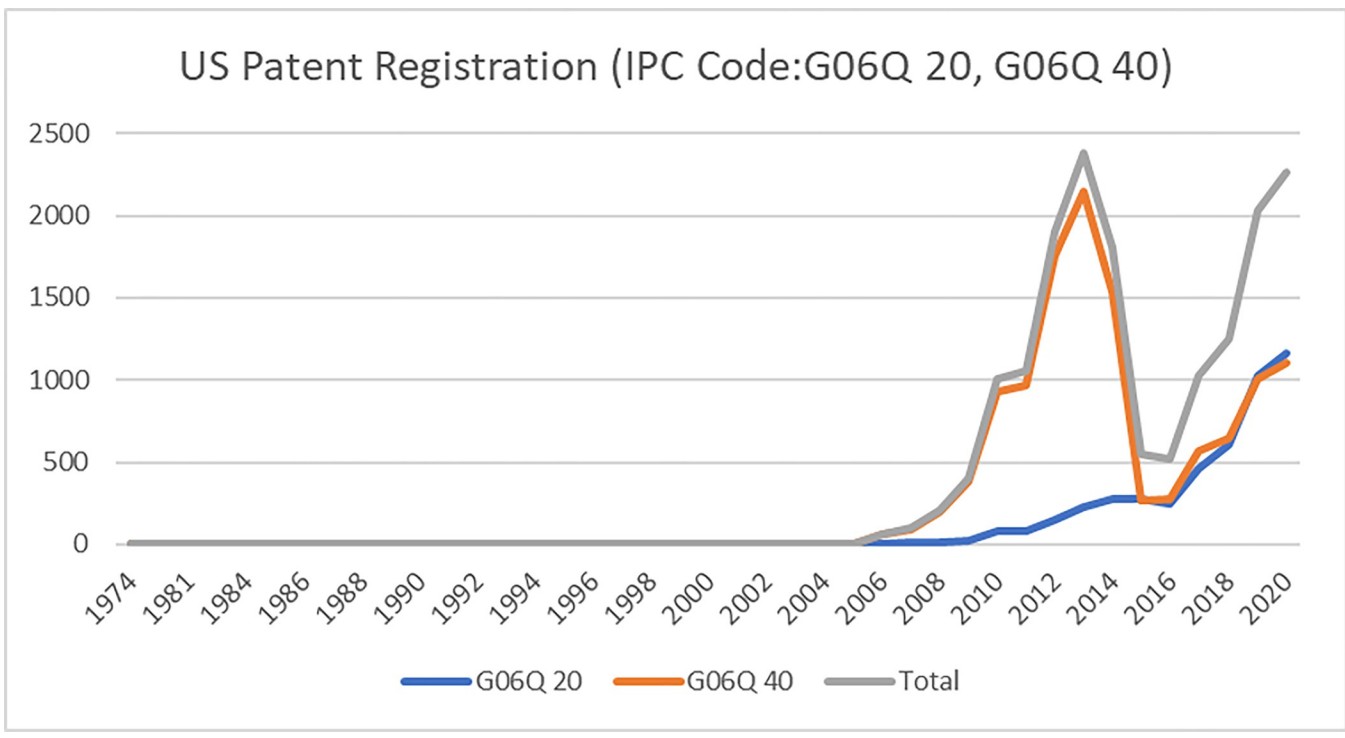

**Fig 2. Distribution of the patents filed at USPTO with IPC codes G06Q 20 or G06Q 40.**

WHAT and HOW Lists and the weight of the HOW and WHAT Lists, QFD is applied to obtain the priority of the HOW list.

*Step 1) Identification of interrelationships*

The core services of commercial banks were divided into five categories, and the keywords that represented them were selected (see **Section 3.1**). If one or more keywords for a specific core service (WHAT List) existed in the patent abstract, that patent was considered a specific "core service"-related patent.

Utilizing the CTM results, we could also identify the probability that each patent refers to a certain technology (HOW List). For example, if we expanded the keywords of a specific core service (WHAT List) matching the patents that were investigated in the study, we could identify the degree of coverage of the core service (WHAT List) in a specific technology (HOW List).

In other words, patents were initially classified into core service-related patent groups, each corresponding to a core service. Subsequently, by evaluating the probability of which technology (HOW List) the patents belong to, we could estimate the degree of interrelationship of the "core service (*i*)" and "technology (*j*)" based on the probabilities of patents belonging to both groups ($p_{ij}^k$).

Finally, we examined how the group is organized by technology by adding the probabilities of all the patents belonging to the group and referring to them as the interrelationship between *i* and *j*, $IR_{ij}$. Specifically, $IR_{ij}$ was obtained by adding $p_{ij}^k$, the probability that patent k belongs to core service *i* and technology *j*, where N is the total number of patents used in this study.

$$IR_{ij} = \sum_{k=1}^{N} p_{ij}^k$$

**Table 2. List of topics extracted from CTM analysis during 2011–2020.**

| Topics No. | Relevant topic words | Topic labels in relation to Technology |
|---|---|---|
| 1 | "may," "user," "information," "device," "include," "vehicle," "methods," "determine," "systems," "associated," "location," "based," "provide," "devices," "insurance," "embodiments," "computing," "autonomous," "various," "platform" | Secure Transaction |
| 2 | "payment," "mobile," "device," "transaction," "merchant," "server," "terminal," "request," "pos," "communication," "point," "sale," "authentication," "wireless," "information," "authorization," "user," "secure," "using," "token" | Mobile Transaction |
| 3 | "card," "account," "credit," "financial," "funds," "system," "payment," "institution," "debit," "prepaid," "customer," "gift," "accounts," "amount," "money," "cards," "merchant," "balance," "bank," "purchase" | Transaction Support Technology |
| 4 | "order," "trading," "price," "market," "trade," "orders," "buy," "trades," "exchange," "bid," "trader," "auction," "prices," "quantity," "time," "offer," "displayed," "sell," "spread," "securities" | Trading Platform |
| 5 | "data," "machine," "reader," "system," "records," "card," "banking," "automated," "check," "read," "processing," "image," "includes," "tax," "operative," "module," "cash," "responsive," "cards," "processor" | ATM (uncontacted financial services) |
| 6 | "rate," "investment," "value," "asset," "portfolio," "contract," "index," "period," "interest," "fund," "loan," "time," "assets," "income," "calculating," "life," "borrower," "return," "adjusted," "annuity" | Financial Product Valuation & Design |
| 7 | "first," "second," "device," "information," "electronic," "server," "user," "terminal," "code," "unit," "display," "communication," "configured," "authentication," "item," "content," "input," "includes," "mode," "network" | Information & Data Processing |
| 8 | "transaction," "account," "method," "one," "request," "associated," "receiving," "least," "merchant," "first," "financial," "includes," "payment," "second," "plurality," "amount," "determining," "identifier," "based," "network" | Personalized Financial Service |
| 9 | "insurance," "one," "risk," "plurality," "based," "policy," "model," "claim," "least," "data," "loss," "coverage," "determining," "set," "premium," "score," "computer," "claims," "vehicle," "tax" | Insurance-related Technology |
| 10 | "services," "service," "digital," "provider," "receipt," "online," "without," "customer," "access," "provides," "party," "internet," "web," "network," "users," "content," "virtual," "transactions," "commerce," "present" | Software for Online Services |
| 11 | "augmented," "activate," "transitory," "housing," "meets," "settings," "contained," "chargeback," "cart," "chip," "sst," "main," "embedded," "reading," "activating," "tablet," "webpage," "medium," "occurred," "confirming" | Hardware Configuration |
| 12 | "augmented," "activate," "accurate," "settings," "activating," "vendors," "analysis," "searching," "marketing," "membership," "co," "activity," "meets," "reality," "webpage," "provisioning," "evaluation," "manufacture," "reserved," "manufacturer" | Data (Information) Analysis |

$p_{ij}^k$ is 0 if patent $k$ is related to neither service i nor technology $j$. The overall summary of the standardized interrelationships, $ST(IR_{ij})$, is displayed in **Table 3** and is derived as follows:

$$ST\left(IR_{ij}\right) = \frac{IR_{ij} - Min(IR_{ij})}{Max(IR_{ij}) - Min(IR_{ij})}$$

*Step 2) Identification of weight for financial services (WHAT) and fintech topics (HOW)*

The weight of **subjectiveness** ($W_{sub}^i$, Weight of WHAT) was determined by five practitioners and experts (three general managers, one IT manager, and one division leader) from the case companies, and the weights were normalized. Pang et al. [50] emphasized the integration

PLOS ONE CTM and QFD analysis: Framework for fintech adoption priority in commercial banks

**Table 3. Results of the standardized interrelationship between service *i* and technology *j*.**

|  | *j* = 1 | 2 | 3 | 4 | 5 | 6 | 7 | 8 | 9 | 10 | 11 | 12 |
|---|---|---|---|---|---|---|---|---|---|---|---|---|
| *i* = 1 | 0.189 | 0.187 | 0.583 | 0.187 | 0.193 | 0.443 | 0.129 | 0.210 | 0.191 | 0.226 | 0.176 | 0.277 |
| 2 | 0.071 | 0.030 | 0.219 | 0.025 | 0.116 | 0.109 | 0.036 | 0.073 | 0.032 | 0.060 | 0.076 | 0.044 |
| 3 | 0.269 | 0.172 | 0.214 | 1.000 | 0.199 | 0.613 | 0.202 | 0.171 | 0.267 | 0.292 | 0.270 | 0.404 |
| 4 | 0.023 | 0.026 | 0.014 | 0.000 | - | 0.031 | 0.018 | 0.007 | 0.002 | 0.035 | 0.010 | 0.016 |
| 5 | 0.122 | 0.169 | 0.364 | 0.077 | 0.259 | 0.148 | 0.171 | 0.221 | 0.074 | 0.169 | 0.146 | 0.100 |

need for subjective weights, such as experts' preferences and determination. The experts were asked to list the investigated financial services in order of importance. Table 4 lists the weights of subjectiveness.

The weight of **importance** ($W_{imp}^j$, the weight of HOW) was computed using citation information, such as the average number of citations and the number of patents for each topic. Specifically, we identified the total number of patents for each fintech topic *j* extracted from the CTM and the number of citations for each of these patents. Subsequently, the average number of citations ($C_{avg}^j$) was obtained by dividing the total number of forward citations by topic by the expected total number of patents with topic j. Forward citation information was obtained using a patented field until July 2021. The standardized interrelationship was derived using the following equation:

$$W_{imp}^j = \frac{C_{avg}^j}{Max(C_{avg})}$$

$$C_{avg}^j = \frac{Total\ number\ of\ Citations^j}{Total\ number\ of\ Patents^j}$$

$$Expected\ total\ number\ of\ Patents^j = \sum_{k=1}^{N} p_j^k$$

$$where\ p_j^k = Probability\ that\ patent\ k\ belongs\ to\ Fintech\ Topic\ j$$

$$Total\ number\ of\ Citations^j = \sum_{k=1}^{N} c^k p_j^k$$

$$where\ c^k = number\ of\ forward\ citation\ of\ patent\ k$$

The overall weight of importance is displayed in **Table 5**.

**Table 4. Weights of subjectiveness.**

| Investigated financial service *i* | Expert 1 | Expert 2 | Expert 3 | Expert 4 | Expert 5 | Standardized weight for subjectiveness ($W_{sub}$) |
|---|---|---|---|---|---|---|
| *i* = 1 | 5 | 5 | 4 | 5 | 4 | 1 |
| 2 | 1 | 2 | 1 | 2 | 1 | 0.2 |
| 3 | 3 | 1 | 3 | 4 | 5 | 0.6 |
| 4 | 2 | 3 | 5 | 1 | 3 | 0.4 |
| 5 | 4 | 5 | 5 | 3 | 2 | 0.8 |

PLOS ONE | https://doi.org/10.1371/journal.pone.0287826 November 1, 2023 10 / 19

Table 5. Weights of importance for each fintech topic *j*.

| Fintech topic *j* | No. of patents | No. of forward citations | $C_{avg}^j$ | $W_{imp}$ |
|---|---|---|---|---|
| 1 | 966 | 55,872 | 58 | 1.000 |
| 2 | 912 | 48,032 | 53 | 0.910 |
| 3 | 917 | 51,309 | 56 | 0.967 |
| 4 | 986 | 42,399 | 43 | 0.743 |
| 5 | 899 | 40,145 | 45 | 0.772 |
| 6 | 956 | 35,184 | 37 | 0.636 |
| 7 | 829 | 39,492 | 48 | 0.824 |
| 8 | 786 | 40,051 | 51 | 0.881 |
| 9 | 814 | 34,931 | 43 | 0.742 |
| 10 | 839 | 41,450 | 49 | 0.854 |
| 11 | 1065 | 44,163 | 41 | 0.717 |
| 12 | 995 | 43,830 | 44 | 0.762 |

The weight of **urgency** ($W_{urg}^j$, the weight of HOW) was calculated using the number of patents related to fintech topics (HOW List) from 2011 to 2020 (**Table 6**). The exponentially weighted moving average (EWMA) was used to assign more weight to the recently applied patents. **Table 6** also shows the EWMA ($\lambda = 0.5$) values by year and the weight of urgency using the EWMA value for the data year 2020.

$$W_{urg}^j = \frac{EWMA_{2020}^j}{Max(EWMA_{2020})}$$

## 4. Results and discussion

### 4.1 Results of QFD analysis

We adopted the QFD with interrelationships and WHAT's weight to obtain a more sophisticated "fintech priority." **Table 7** shows the results of the QFD with interrelationships and weights applied.

Topics 3 (transaction support technology), 1 (secure transactions), and 4 (trading platforms) were evaluated as the top three high priorities in the QFD analysis. By contrast, Topics 9 (insurance-related tech), 6 (financial product valuation and design), and 11 (hardware

Table 6. Number of patents by year for each fintech topic *j*.

| | 2011 | 2012 | 2013 | 2014 | 2015 | 2016 | 2017 | 2018 | 2019 | 2020 | Sum | $EWMA_{2020}$ | $W_{urg}$ |
|---|---|---|---|---|---|---|---|---|---|---|---|---|---|
| *j* = 1 | 55 | 89 | 112 | 88 | 38 | 36 | 81 | 100 | 184 | 183 | 966 | 158.38 | 1.000 |
| 2 | 44 | 79 | 102 | 85 | 47 | 39 | 87 | 99 | 165 | 165 | 912 | 145.01 | 0.916 |
| 3 | 86 | 138 | 152 | 128 | 33 | 32 | 62 | 66 | 115 | 104 | 917 | 96.32 | 0.608 |
| 4 | 120 | 190 | 171 | 125 | 25 | 21 | 58 | 63 | 112 | 101 | 986 | 93.44 | 0.590 |
| 5 | 80 | 115 | 131 | 94 | 48 | 38 | 62 | 68 | 131 | 133 | 899 | 115.04 | 0.726 |
| 6 | 136 | 200 | 203 | 132 | 22 | 21 | 37 | 45 | 79 | 81 | 956 | 71.58 | 0.452 |
| 7 | 46 | 83 | 100 | 80 | 36 | 31 | 66 | 76 | 143 | 167 | 829 | 135.72 | 0.857 |
| 8 | 51 | 93 | 113 | 95 | 28 | 25 | 52 | 67 | 128 | 133 | 786 | 112.77 | 0.712 |
| 9 | 80 | 113 | 133 | 99 | 25 | 21 | 44 | 60 | 124 | 114 | 814 | 101.17 | 0.639 |
| 10 | 67 | 111 | 134 | 104 | 39 | 31 | 65 | 69 | 109 | 111 | 839 | 98.54 | 0.622 |
| 11 | 96 | 139 | 157 | 116 | 53 | 51 | 80 | 78 | 146 | 148 | 1,065 | 129.63 | 0.818 |
| 12 | 112 | 159 | 198 | 128 | 28 | 25 | 56 | 66 | 112 | 112 | 995 | 99.39 | 0.628 |

**Table 7. Results of the QFD for financial services using standardized interrelationship with weights.**

| | j = 1 | 2 | 3 | 4 | 5 | 6 | 7 | 8 | 9 | 10 | 11 | 12 | $W_{sub}$ |
|---|---|---|---|---|---|---|---|---|---|---|---|---|---|
| i = 1 | 0.189 | 0.187 | 0.583 | 0.187 | 0.193 | 0.443 | 0.129 | 0.210 | 0.191 | 0.226 | 0.176 | 0.277 | 1.000 |
| 2 | 0.071 | 0.030 | 0.219 | 0.025 | 0.116 | 0.109 | 0.036 | 0.073 | 0.032 | 0.060 | 0.076 | 0.044 | 0.200 |
| 3 | 0.269 | 0.172 | 0.214 | 1.000 | 0.199 | 0.613 | 0.202 | 0.171 | 0.267 | 0.292 | 0.270 | 0.404 | 0.600 |
| 4 | 0.023 | 0.026 | 0.014 | 0.000 | - | 0.031 | 0.018 | 0.007 | 0.002 | 0.035 | 0.010 | 0.016 | 0.400 |
| 5 | 0.122 | 0.169 | 0.364 | 0.077 | 0.259 | 0.148 | 0.171 | 0.221 | 0.074 | 0.169 | 0.146 | 0.100 | 0.800 |
| $W_{imp}$ | 1.000 | 0.916 | 0.608 | 0.590 | 0.726 | 0.452 | 0.857 | 0.712 | 0.639 | 0.622 | 0.818 | 0.628 | |
| $W_{urg}$ | 1.000 | 0.910 | 0.967 | 0.743 | 0.772 | 0.636 | 0.824 | 0.881 | 0.742 | 0.854 | 0.717 | 0.762 | |
| Priority | 0.472 | 0.367 | 0.619 | 0.374 | 0.304 | 0.277 | 0.283 | 0.318 | 0.198 | 0.299 | 0.278 | 0.294 | |
| Rank | 2 | 4 | 1 | 3 | 6 | 11 | 9 | 5 | 12 | 7 | 10 | 8 | |

configuration) were assigned low priorities. Consequently, Topics 3, 1, and 4 were considered important technologies that needed to be acquired in a timely manner.

Notably, for some specific fintech topics, there is a noticeable difference in the priority order when only subjective weight ($W_{sub}$) is applied compared with when both subjective weight ($W_{sub}$) and urgency weight ($W_{urg}$) are applied. For example, in the case of Topic 6 (financial product valuation and design), when only the subjective weight was applied, it was derived as the second-most important topic, but when both the subjective and urgency weights were applied, its priority was reclassified as the second-least important topic.

Conversely, in the case of Topics 1 (secure transaction) and 2 (mobile transaction), when only the subjective weight was applied, they were derived as being ninth and tenth in importance, respectively; however, when both subjective and urgency weights were considered, they were reclassified as the second- and fourth-most important topics. These results suggest that if technology priorities are derived by reflecting only the opinions of experts from commercial banks, the findings may not capture a holistic view of managerial and technical trends. The proposed methodology should include comprehensive inputs from practitioners and development trends of fintech technologies.

## 4.2 Investment strategy for commercial banks

As recently noted by Brandl and Hornuf [26], traditional and commercial banks can utilize various investment strategies to achieve digitalized financial services, such as full integration of another company, strategic partnerships between firms, and spin-off operations initiated by banks. This study provides complementary decision support guidelines that practitioners can use for investment decisions and partnership approaches for specific technologies. Specifically, practitioners can utilize the weights assigned by experts and patent information to make integrative decisions regarding technology acquisition.

Fig 3 presents the potential classification and relevant strategic guidelines for each technology topic. For example, Group I may be considered for full integration or strategic partnerships among firms or competitors. If patent owners are emerging technology or start-up fintech companies that seek to merge opportunities, the traditional bank may benefit from the consideration of full acquisition. However, if patent owners are large national companies (e.g., Bank of America) or cannot be considered for acquisition for geopolitical reasons, then the strategic partnership may be a more intuitive choice.

For Group II technologies, integrative spin-off operations with internal and external development efforts are needed. Internal development offers various benefits in fostering dynamic

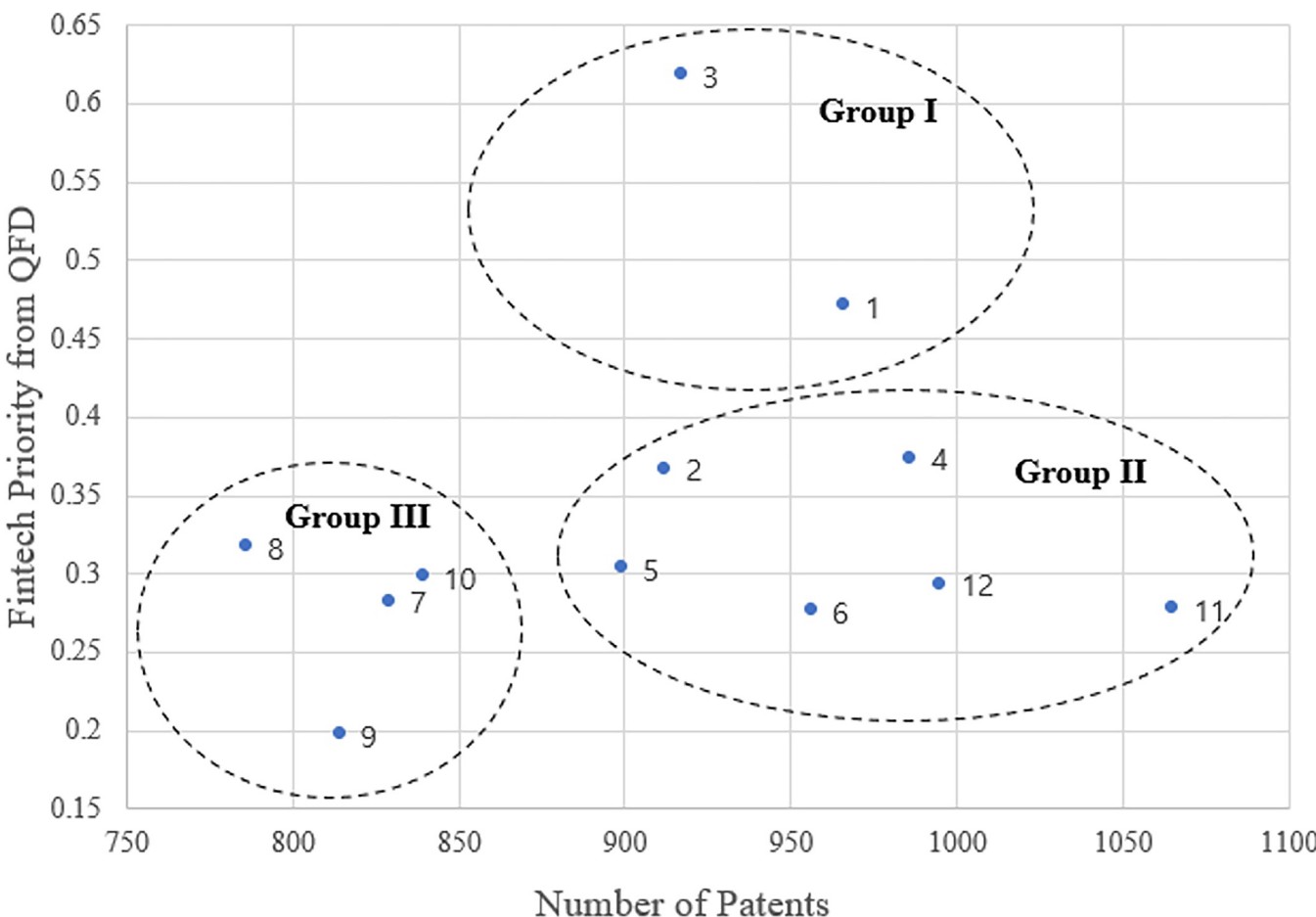

**Fig 3. Strategic guidelines for fintech investment based on the number of patents (*x*-axis) and fintech priority (*y*-axis).**

capabilities related to digital servitization, but also requires consideration of external factors, such as environmental contingencies [35]. Based on the extent of patents that exist for these technology topics, a joint collaboration between banks and fintech firms can create a fine-tuned service, thereby providing synergistic performance to satisfy market needs [51].

Finally, for Group III it is necessary to adopt conservative approaches in both investment and development strategies. Strategic partnership is recommended for the technology topics under this group, which will require a high level of collaboration for internal and external technology development for a desirable investment outcome. Results of the grouping and its implications are summarized in **Table 8**. We further interpret the abovementioned findings in the context of commercial banks in South Korea.

## 5. Conclusion

This study primarily aimed to resolve the absence of a decision framework for investment in technology, specifically from the perspective of commercial banks. Rooted in the technology adoption theory, customer-centric financial services were classified based on the integrative views of the literature and field practitioners. Then, the emerging topics of technological trends were identified using a patent database. CTM and QFD analyses were applied to extract technology investment priorities and recommendations for acquisition strategies for each fintech topic.

**Table 8. Examples of degree of investment based on the partnership.**

| | Full integration | Spin-off operations | Strategic partnership |
|---|---|---|---|
| Classification (Examples) | Group I (Topics 1, 3) | Group II (Topics 2, 4, 5, 6, 11, 12) | Group III (Topics 7, 8, 9, 10) |
| Technology details | Transaction security, Transaction support technology | Mobile transactions, trading platform, uncontacted financial service, Financial product valuation and design, hardware configuration, data analysis | Information and data processing, Personalized financial services, Insurance-related technology, Software for online services |
| Fintech service priority level (y-axis) | Moderate to High | Low to Moderate | Low |
| Fintech development status (x-axis) | Moderate to High | Moderate to High | Low |
| Development strategy characteristics | Time-sensitive involvements are required in both technology development and service integration as an immediate integration approach | The readiness of technology is considered relatively low. Instead of immediate application, the feasibility and assessments are recommended under collaborative efforts | Collaboration for both internal and external technology developments are recommended for a synergistic performance |
| Degree of collaboration | High | Moderate | Low |
| Degree of investments | High | Moderate | Low |
| Examples in the South Korean context | Leading commercial bank in South Korea established Global Loyalty Network ("GLN"), which is a globally integrated platform that offers cross-border use of digital assets | Leading commercial bank in South Korea established a joint venture (JV) with medium size Tech company to lead the small and medium-sized business (SME) market by connecting financial ICT and technological competitiveness | Most large commercial banks in South Korea are incubating several technology startups for future growth |

This study has significant implications according to the findings. First, technology investment priorities must be determined based on the overall financial service strategy. R&D and technology investments are likely to lead to superior innovation capabilities which can thereby enhance new core competency based on the theory of dynamic capabilities [36,44]. However, the recent trends in prioritizing technology investment have mainly investigated the investment productivity utilizing efficiency measure approaches such as data envelopment analysis [52]. Our findings instead highlight the importance of the identification of service strategy prior to deciding the investment priorities as the results are differentiated based on the urgency of service development. Banks must decide on appropriate investment strategies and develop internal and external resources instead of depending on the technology advancement for the investment decision.

Second, our findings reshape technology acquisition strategies of commercial banks. Insights obtained from our study includes (1) transaction support technology, secure transactions, and trading platforms are commonly evaluated as the most critical technology topics; and (2) commercial banks are recommended to make investment strategies such as M&As, strategic partnerships, and spin-off operations considering the number of patents, importance of technology, and size of patent-owned fintech firms. Despite various literatures emphasizing the importance of technology-empowered personalized services [53], our study well aligns with Cao et al.'s [21] study in that transaction security is the utmost priority. This may be due to the high level of collaboration required for security technology. Thus, the proposed approaches is novel in comparison to the recent trend in emphasis of the fintech adoption (without consideration of the investment orders), our approach demonstrates the practical ways to reflect both.

Lastly, the alignment between financial service development and fintech trends can be improved through the decision support framework proposed in this study. Most importantly,

this framework can be generalized and applied to other firms. The outcomes of this research approach can support and enable practitioners to make strategic decisions to enhance the productivity of fintech applications in meeting financial services. To the best of our knowledge, this is the first study to use a QFD structure to identify fintech investment orders using objective technological trends based on patent information and subjective financial

It should be noted that without support from government regulations and policies, it is difficult for companies to achieve financial innovation or make sustainable investment decisions. When the Financial Services Commissions in South Korea proposed supportive policies, such as the Special Act on Support for Financial Innovation in 2019 and the Electronic Financial Transaction Act in 2017, various companies, including big tech and e-commerce companies, began various investment activities with a positive outlook at the entrance into the financial industry. This appears to be well aligned with the global trend in fostering the ecosystem of fintech startups and commercial banks in the financial sector [54].

However, a vague regulatory and policy stance remains regarding developing comprehensive and advanced payment and settlement services. For example, the regulatory system is currently ambiguous regarding allowing fintech firms to acquire and handle personal data or restricting them to traditional banks only. Furthermore, regarding virtual asset-related businesses, the current South Korean government maintains a conservative view and stands by a policy prohibiting financial services from engaging in transactions involving virtual assets. For example, the recently announced Act on Reporting and Using Specified Financial Transaction Information requires individuals or firms to report cryptocurrency-related transactions.

Based on the overall trend of fintech development and the involvement of various companies, regulators and policymakers need to actively consider how to support effective collaboration with appropriate incentive schemes. This can be further explained by observing the top 20 firms that own patents for Topics 3 and 4 (see **S1 Appendix**). Some patents related to Topic 3 are owned by global financial companies, whereas others are owned by individuals or small-sized companies. As most patents related to Topic 4 are owned by large exchanges and global financial firms, it is difficult to acquire technology through M&A. Therefore, to introduce and develop Topic 4-related technologies, it is necessary to consider paying a fee and forming a technology alliance.

Globally, particularly in South Korea, policymakers tend to separate the management of financial and nonfinancial corporations. Specifically, in the case of commercial banks, various direct or indirect regulations tend to disturb the rapid introduction of fintech by commercial banks. Most commercial banks worldwide do not speed up digital transformation because of regulations. For the long-term development of the financial system, it is necessary to quickly introduce developing fintech technology into traditional commercial banks, and active support from policymakers is required.

While this study provides commercial banks' acquisition strategy for fintech, it also intends to stimulate greater interest in understanding fintech applications from the perspective of traditional or commercial banks. To this end, this study proposes three research avenues to foster synergistic collaboration for greater performance in the financial industry. First, the classification of financial services and their subjective importance can benefit from inputs from other regions. While this study provides an integrative perspective of the literature and field practitioners from South Korea, it does not provide as comprehensive a perspective as the patent database. Second, the QFD framework with several subjective perspectives can be evaluated and verified in different contexts. For example, the framework's validity in resolving the potential gap between several subjective views can be further investigated from a decision-support system perspective, as other complementary approaches can indefinitely support practitioners in making better investment decisions. Third, the time that takes until commercialization

from patent application should be considered when making strategic planning for the banks. For example, Broekel [55] and Daiha [56] noted a time lag for patent applications to come into effect. Finally, internal and external factors in fintech application-based financial services should be considered for long-term planning. For example, certain technologies may have higher volatility in terms of providing stable technologies and services to end customers. For sustainable investment planning and strategy, both market readiness and technological uncertainty should be incorporated in the decision-making process. These areas are left for further studies.

## Supporting information

**S1 Appendix. List of patents with the highest relation percentage for Topics 1 through 12.** (DOCX)

## Author Contributions

**Conceptualization:** Donghoon Kang, So Young Sohn.

**Data curation:** Donghoon Kang.

**Formal analysis:** Donghoon Kang.

**Funding acquisition:** So Young Sohn.

**Investigation:** Donghoon Kang.

**Methodology:** Donghoon Kang, So Young Sohn.

**Supervision:** So Young Sohn.

**Writing – original draft:** Donghoon Kang.

**Writing – review & editing:** So Young Sohn.

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
