## [Decision Letter · Decision Letter 0]

19 Jan 2023

PONE-D-22-32567Patent-based contextual topic modeling and QFD analyses for commercial banks’ adoption priority for fintechPLOS ONE

Dear Dr. Sohn,

Thank you for submitting your manuscript to PLOS ONE. After careful consideration, we feel that it has merit but does not fully meet PLOS ONE’s publication criteria as it currently stands. Therefore, we invite you to submit a revised version of the manuscript that addresses the points raised during the review process.

Though a reviewer rejected the manuscript, the reviewer provided many valuable and constructive comments. Considering five reviewers' useful comments and the interesting topic of the manuscript, I would like to give you a chance to revise your manuscript during the special period. The revised manuscript will undergo the next round of review by the same reviewers.

We look forward to receiving your revised manuscript.

Kind regards,

Baogui Xin, Ph.D.

Academic Editor

PLOS ONE

Journal Requirements:

"The corresponding author received the National Research Foundation of Korea (NRF) grant funded by the Korea government (MSIT) [grant number 2020R1A2C2005026]"

Reviewers' comments:

Reviewer's Responses to Questions

**Comments to the Author**

1. Is the manuscript technically sound, and do the data support the conclusions?

Reviewer #1: Yes

Reviewer #2: Yes

Reviewer #3: No

Reviewer #4: Partly

Reviewer #5: Partly

2. Has the statistical analysis been performed appropriately and rigorously? 

Reviewer #1: Yes

Reviewer #2: Yes

Reviewer #3: No

Reviewer #4: Yes

Reviewer #5: Yes

3. Have the authors made all data underlying the findings in their manuscript fully available?

Reviewer #1: Yes

Reviewer #2: Yes

Reviewer #3: No

Reviewer #4: No

Reviewer #5: Yes

4. Is the manuscript presented in an intelligible fashion and written in standard English?

Reviewer #1: Yes

Reviewer #2: Yes

Reviewer #3: Yes

Reviewer #4: Yes

Reviewer #5: Yes

5. Review Comments to the Author

Reviewer #1: Manuscript Number: PONE-D-22-32567

Title: Patent-based contextual topic modeling and QFD analyses for commercial banks’ adoption priority for fintech This study proposes a decision-making support framework for banks by considering both aspects of the fintech push of related emerging technologies and the market pull of banking services.

Abstract: The abstract should sequence the objectives, techniques, and concise results. Revise accordingly

Introduction: In the introduction section, clear research objectives are missing. This section should clarify the background, objectives, research gaps, and innovations. Be concise with the authors' research problem, gaps, objectives, and innovation contribution.

Literature review: Add more appropriate and advanced literature on commercial banking. You can get the idea from the following studies. http://dx.doi.org/10.2139/ssrn.3107770

https://doi.org/10.1371/journal.pone.0270406

Research methodology and data analysis

This study employs QFD methodology to derive “fintech priority” based on finance-related patent data and opinions collected from commercial bank practitioners. Methods are advanced and accurate. Add some detail that could justify the application of these methods in the study.

Clarify the Sources of variable selection in table 1.

Discussion Add more references to interpret the results in sections 4.1, 4.2, and 4.3. Further, elaborate on the results in detail.

Conclusion and future studies. Add limitations and future research ideas in the conclusion section. Give more policy implications for commercial banks.

Avoid grammatical and typo errors and revise the manuscripts for these concerns.

Improve the quality of Figure 1.

Reviewer #2: Title: Patent-based contextual topic modeling and QFD analyses for commercial banks’ adoption priority for fintech, Manuscript Number: PONE-D-22-32567

This study unfolds the innovative framework for commercial banks in South Korea. It needs comprehensive improvement to be accepted for publication. Hopefully, my detailed comments will help the authors improve the manuscript's quality.

Abstract: This section is well organized; however, it must avoid grammatical and typo mistakes.

Introduction: The problem statement and innovations need to be clearer and more detailed.

Literature review: Technological factor impact on commercial banks should be discussed more in the literature section. Add more advanced literature on the topic.

Research methodology and data analysis. This section is properly organized. But still contain grammatical and typo mistakes. Revise it

Discussion More citations are required to back your study results in the discussion section. Find appropriate literature to strengthen your results.

Conclusion and future studies. Revise the conclusion to describe your study's problem, methods, and innovative findings. Further research topics related to this framework could be discussed here.

Reviewer #3: This paper attempts to elaborate the priority of financial technology that commercial banks should pay attention to. The structure is relatively clear, but there are some serious deficiencies. In particular, the scientific soundness of the paper needs to be further demonstrated.

1.The practical significance of the paper is poor. Whether financial practitioners, regulators or scholars, it is generally agreed that financial security is the most important issue in the financial industry, and the contribution of FinTech to this issue is mainly reflected in transaction technology and information processing capabilities. The authors seem to have just clarified a common sense through complicated means and a lot of work, which results in the lack of valuable policy implications in the paper.

2.Another fatal deficiency is why the US patent data should be used, rather than that of South Korea (if any). This leads to all your conclusions may be imprecise. The authors need to explain the relationship between American technology and South Korean FinTech applications or the relationship between American patent data and South Korean banking.

3.Are the research objects selected by the authors broadly representative? As far as the world's major economies are concerned, banking is a competitive industry, and banks of different sizes often have different business strategies. Therefore, the authors only select the four largest banks in South Korea, and the rationality of this need to be fully demonstrated.

4.Please notes that the introduction section plays an important role in attracting attention. When the introduction is bad, the reviewer is likely to reject the paper. As a Research Article, the authors should particularly emphasize the contribution of this research to the body of knowledge. Unfortunately, I have not seen the relevant discussion.

5.In Section 4.2, the authors need to make a reasonable explanation for the grouping of topics. Why are they grouped in this way? What is the basis for dividing them into these three groups? Is it reasonable? Of course, the author did not explain why commercial banks should adopt different investment strategies for different groups.

6.The abstract is unclear. The author needs to sort out the logic and expression of the abstract.

7.I did not see the relevant data in the Supporting Information files.

Other minor deficiencies.

1.In line 201 of the PDF document, “These firms have approximately over 305 billion USD (400 trillion KRW) with more than 25,000 employees.”, where “305 billion USD” is a loan, deposit or total asset?

2.The authors lack the necessary explanation of the content to be proved by the figures and tables.

Reviewer #4: This paper proposes a decision-making support framework for banks by considering both aspects of the fintech push of related emerging technologies and the market pull of banking services. The subject is very important and it's currently something of a "hot topic", but there are some problems need to be revised.

#1. The author needs to polish their manuscript in language.

#2. The current innovations are not convincing. They are more likely to express the functions. It is suggested to rewrite the innovation point.

#3. As the QFD methodology has been mentioned many times in this paper, it may be better to place the introduction of QFD in the front of Chapter 3.

#4. This paper takes the four of the largest banks as examples, they are all excellent companies. Thus, how to convince readers the proposed framework can be applied to other financial service firms.

#5. If possible, the overall length of the article can be reduced without affecting the expression of the author's core ideas.

Reviewer #5: COMMENTS TO THE AUTHOR

1. The topic is too complex for global readers understanding. It needs to be updated. I suggest the topic to read as CTM and QFD analysis: Framework for Fintech Adoption Priority in Commercial banks.

2. Abstract:

Bring out the purpose/objective of the study in the abstract. It’s missing. The abstract need to be reorganized bringing out the purpose of the study, methodology, finding and the value of the study. There seems to be a mixture up of issues. Give a summary of the study in the abstract to bring out the entire picture of the study in a summarized form.

3. In the introduction:

(i) Show a global perspective of the challenges facing banks in the choice and adoption of fintech. This will enrich the background bringing out the problem in a global context before you narrow down to Kerean banks.

(ii) The proposed use of QFD framework in the introduction is in appropriate, instead it should be brought up in the methodology section in a systematic manner.

(iii) Create a section to discuss commercial banks in Korea in terms of their choice of fin-tech and adoption of the technology. This will help the readers I the global context understand the status of Korean banks in terms of fintech adoption.

4. In Literature review:

(i) Section 2.1 discusses mostly the challenges facing commercial banks . This section could fit well in the introduction section to aid in bringing up the problem under investigation. This section also concentrates more on discussing the benefits of fintech hence excluding the choices and strategies for adoption of fintech which are two major components of this study.

(iv) Theoretical literature missing. Introduce a section on theoretical review. Introduce a theory that guides this study.

(v) Introduce a section on empirical literature and review studies relating to choice of fintech adoption in commercial banks, strategies of on the choice of fintech adopted, effect of patent based technologies and their effect. This will help in identifying research gaps.

5. In the methodology:

(i) Show the target population, how many commercial banks in Korea and justify the choice of the sample selected. Bring out the criteria used for selecting four firms for this study. Are they not operating under the same environment, have they not adopted the fin tech or what? I suggest that such a study is relevant if all firms are involved in the study sample unless there is a major reason for them to be excluded.

(ii) The study uses patents filed at USPTO up to the year 2020. There might be some more patents that could affect the choice of fintech adoption to date. Therefore ensure that the study is updated to capture patents filed up to date to make the study more relevant with current trends.

(iii) To determine the weight of subjectiveness, brig out the criteria that was used to select the five practitioners/experts in each firm. Why choose five experts, is there any reason for this number?

(iv) In some section you claim to use 10 patents, in other section you use more patents in computation of weight of importance. There seems to be a disconnect between the two hence a clear justification id needed.

(v) In the methodology section you seem to have brought up the results. Concentrate on bringing out methodology used in section 3. The result fits well in section 4 where they are presented together with their discussion and interpretation.

6. Discussion

(i) Discussion was ok, however there is no link to the reviewed literature. We should be able to see whether, the outcome of the study support or contradict prior empirical evidence.

7. Conclusion should be done before implications to enhance the flow of ideas.

8. Check the entire document for typo and grammatical errors.

6. PLOS authors have the option to publish the peer review history of their article (what does this mean?). If published, this will include your full peer review and any attached files.

Reviewer #1: No

Reviewer #2: No

Reviewer #3: No

Reviewer #4: No

Reviewer #5: No

---

## [Author Response · Author response to Decision Letter 0]

1 Apr 2023

I updated "Response to Reviewer" file on your system.

---

## [Decision Letter · Decision Letter 1]

7 May 2023

PONE-D-22-32567R1CTM and QFD analysis: Framework for Fintech Adoption Priority in Commercial banksPLOS ONE

Dear Dr. Sohn,

Thank you for submitting your manuscript to PLOS ONE. After careful consideration, we feel that it has merit but does not fully meet PLOS ONE’s publication criteria as it currently stands. Therefore, we invite you to submit a revised version of the manuscript that addresses the points raised during the review process.

We recommend that it should be revised taking into account the changes requested by the reviewer. Since the requested changes includes Minor Revision, the revised manuscript will undergo the next round of review by the same reviewers or only by the Academic Editor.

We look forward to receiving your revised manuscript.

Kind regards,

Baogui Xin, Ph.D.

Academic Editor

PLOS ONE

Journal Requirements:

Reviewers' comments:

Reviewer's Responses to Questions

**Comments to the Author**

1. If the authors have adequately addressed your comments raised in a previous round of review and you feel that this manuscript is now acceptable for publication, you may indicate that here to bypass the “Comments to the Author” section, enter your conflict of interest statement in the “Confidential to Editor” section, and submit your "Accept" recommendation.

Reviewer #1: All comments have been addressed

Reviewer #2: All comments have been addressed

Reviewer #4: All comments have been addressed

Reviewer #5: (No Response)

2. Is the manuscript technically sound, and do the data support the conclusions?

Reviewer #1: Yes

Reviewer #2: (No Response)

Reviewer #4: Yes

Reviewer #5: Yes

3. Has the statistical analysis been performed appropriately and rigorously? 

Reviewer #1: Yes

Reviewer #2: Yes

Reviewer #4: Yes

Reviewer #5: Yes

4. Have the authors made all data underlying the findings in their manuscript fully available?

Reviewer #1: Yes

Reviewer #2: Yes

Reviewer #4: Yes

Reviewer #5: Yes

5. Is the manuscript presented in an intelligible fashion and written in standard English?

Reviewer #1: Yes

Reviewer #2: Yes

Reviewer #4: Yes

Reviewer #5: Yes

6. Review Comments to the Author

Reviewer #1: (No Response)

Reviewer #2: (No Response)

Reviewer #4: The author has answered all questions and made detailed revisions to the article. I agree to accept this paper.

Reviewer #5: COMMENT TO THE AUTHOR

1. The topic was updated as suggested in the first review.

2. Abstract was updated to bring out the purpose of the study and the methodology.

3. INTRODUCTION

(i) A global perspective of the challenges facing banks in the choice and adoption of fintech has been incorporated as suggested previously hence the background has been enriched bringing out the problem in a global context.

(ii) Challenges facing Korean banks has not been discussed in the background in terms of fintech adoption. This is very vital for this study.

(iii) I suggest the Author to create a section in the background to discuss commercial banks in Korea in terms of their choice of fin-tech and adoption of the technology. This parts has not been done as was suggested previously. hence with the current version in the introduction, the reader is not able to conceptualize the situation of fintech adoption in the context of Korean Banks.

LITERATURE

1. The empirical literature was well updated to capture the choice of fintech in banks and the strategies adopted.

2. However, the Author has not indicated the theories that this study is anchored to. this is very vital. This was previously suggested. Let the Author briefly relate this study to an existing theory by including a section on theoretical literature.

METHODOLOGY:

1. As previously suggested, the author ought to have indicated the number of commercial banks operating in south Korea, and further justify the choice of the four selected. as it is in this revised version, the reader is not able to understand the choice of the four banks well. as a matter of fact, all banks operating in a given country are regulated by the central banks in that country as far as regulations and operations is concerned. it would have been appropriate to show the target population and how the sample was arrived at.

2. As previously suggested, the Author should separate methodology and results. Bring out methodology clearly. let the results be in a different section. may be you can rename section 4 as Results and Discussion.

DISCUSSION

1. As previously suggested, let the discussion be linked to the literature. This has not been done.

7. PLOS authors have the option to publish the peer review history of their article (what does this mean?). If published, this will include your full peer review and any attached files.

Reviewer #1: No

Reviewer #2: No

Reviewer #4: No

Reviewer #5: No

---

## [Author Response · Author response to Decision Letter 1]

12 Jun 2023

Please check "Response to Reviewer_20230612" file.

---

## [Editor Report · Decision Letter 2]

15 Jun 2023

CTM and QFD analysis: Framework for Fintech Adoption Priority in Commercial banks

PONE-D-22-32567R2

Dear Dr. Sohn,

We’re pleased to inform you that your manuscript has been judged scientifically suitable for publication and will be formally accepted for publication once it meets all outstanding technical requirements.

Kind regards,

Baogui Xin, Ph.D.

Academic Editor

PLOS ONE
---

## [Editor Report · Acceptance letter]

19 Jun 2023

PONE-D-22-32567R2 

CTM and QFD analysis: Framework for Fintech Adoption Priority in Commercial banks 

Dear Dr. Sohn:

I'm pleased to inform you that your manuscript has been deemed suitable for publication in PLOS ONE. Congratulations! Your manuscript is now with our production department. 

Kind regards, 

on behalf of

Professor Baogui Xin 

Academic Editor

PLOS ONE